# Earliest Deadline First Scheduling for Real-Time Computing in Sustainable Sensors

**Maryline Chetto** [1,*] and **Rola El Osta** [2]

1 LS2N Laboratory-UMR CNRS 6004, Nantes Université, CEDEX 03, 44321 Nantes, France
2 LENS Laboratory, Lebanese University, Saida P.O. Box 813, Lebanon
* Correspondence: maryline.chetto@univ-nantes.fr

**Abstract:** Energy harvesting is a green technology that authorizes small electronic devices to be supplied for perpetual operation. It enables wireless sensors to be integrated in applications that previously were not feasible with conventional battery-powered designs. Intermittent computing and scheduling are the two central aspects of designing a Real-Time Energy Harvesting (RTEH) sensor, generally used to monitor a mission critical process. Traditional scheduling algorithms fail to timely execute the hard deadline tasks because they accommodate no fluctuations in power supply and therefore no intermittent computing. A suitable energy-harvesting-aware scheduling algorithm has been proposed so as to achieve a higher schedulability rate. Unlike the classical EDF (Earliest Deadline First) scheduler, the ED-H algorithm is idling and clairvoyant, with an improved performance in terms of the deadline missing ratio. This paper reviews the main advances in dynamic priority scheduling based on EDF for energy-neutral systems.

**Keywords:** sustainable sensor; real-time computing; energy harvesting; energy neutrality; preemptive scheduling; earliest deadline first





## 1. Introduction

The new generation of low-power sensor nodes allows local, remote and autonomous control of a very large range of mission critical products. Their use in Cyber–Physical Systems (CPS) tends to spread quickly in automotive systems, appliances, military and security systems, etc. [1,2]. Wireless sensors constitute the key parts for a vast range of computing and communication infrastructures in global markets [3–6]. Energy harvesting (EH) is a technology that enables a small standalone sensor to function perpetually (more specifically, the lifespan in the order of one or several decades) and continuously, without needing a power-line connection or battery replacement. For this reason, EH technology has become quite popular. Thanks to the developments in ultra-low-power semiconductors, EH will facilitate exciting classes of new embedded system applications [7–10]. It is expected to become more prevalent in the near future thanks to the numerous benefits it provides to the embedded system designs. Nonetheless, a complete EH-powered sensor system uses a lot of various components (see [11] for a review). Figure 1 depicts the framework of an energy-harvesting system as the power-source element of a typical low-power sensor.

Different types of energy can be captured from the environment. Outdoor light and indoor light are the most famous ones [12]. This energy can be converted into electricity through photoelectric cells. Some objects may produce mechanical energy. When they vibrate or move, they can produce electricity since vibrations generate considerable voltage when applied to a piezoelectric material [13–16]. Thermoelectric energy is also a source of possible energy for sensor nodes [17–19]. Importantly, ambient energy most often is unstable and intermittent [20]. Energy-harvesting products have to operate continuously. Consequently, they must incorporate a rechargeable energy storage device and power-control circuitry [21,22]. When designing and developing standalone sensor systems,

engineers have to address a lot of design issues relative to the performance characteristics of the sensor, the capacity of the energy storage unit, the processing performance of the microcontroller, etc.

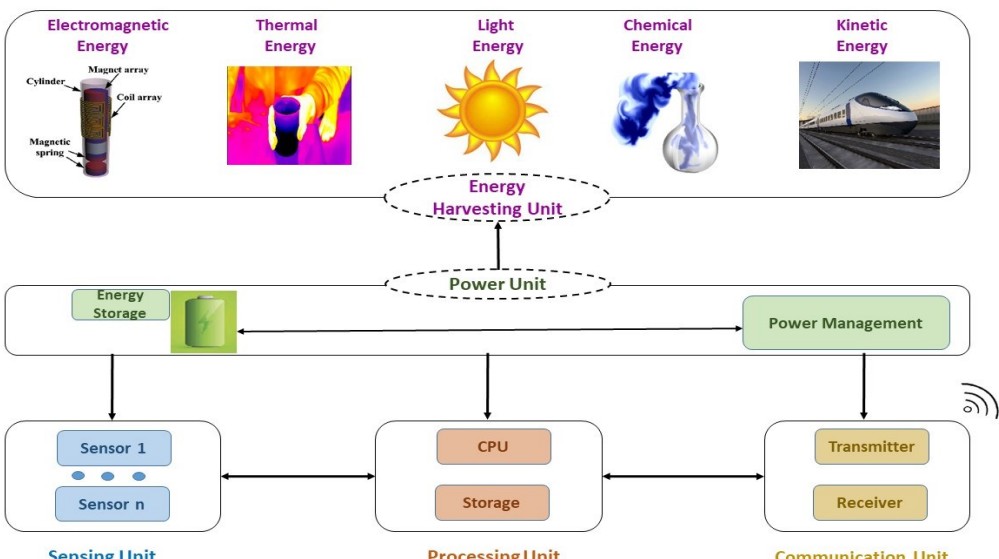

**Figure 1.** A typical real-time energy-harvesting sensor node.

The implementation of energy-harvesting systems is rapidly changing as better power management chips are introduced and harvesters are improved [23,24]. Electronic engineers gain application experience with EH technology. Choosing the right microcontroller is a central step for a very low power embedded device.

A good microcontroller will have several power-down modes so as to minimize power consumption. It will perform fast processing so as to satisfy the timing requirements of the real-time programs, and the extremely fast wake-up times from power-down modes. Electronic circuits must spend as much time as possible in a low power state before switching to the active operating mode [25].

When the engineer incorporates EH technology into a sensor, he has to consider a power-management function so as to handle fluctuations in the power generated by the harvester [26,27]. In most energy-harvesting devices, an energy-storage device serves as a buffer energy unit also called a reservoir between the load (i.e., the different software tasks in execution on the microcontroller) and the energy harvester (i.e., a solar panel, for example). The battery or supercapacitor provides power to the electronics when the harvester cannot produce power or when the computing device requires more current than the harvester can provide.

These specific characteristics of sensor-based systems impose additional challenges on Operating Systems (OS) because their role and requirements deviate from a traditional OS design [28]. The role of the OS is primarily to act as a resource manager. Typically, resources include the processing unit and other hardware such as memories, timers, etc. The OS has to provide allocation of these resources, taking into account the constraints and performance criteria. In a real-time system, the application tasks invoke the different OS services through system calls. The tasks' access the resources according to their priority, which generally reflects an urgency and/or criticality. Scheduling determines the order in which tasks are executed on the processing unit [29,30]. In traditional real-time computer systems, the goal of a scheduler is to ensure feasibility, i.e., to optimize the ratio of deadline missing. In all RTOSes (Real Time Operating Systems), the scheduler can accommodate the timing requirements by assigning fixed or dynamic priority to each task, by authorizing pre-emptions. In most cases, sensor-based systems are utilized to continuously monitor/control a given phenomenon and to process data measurements such as temperature, pressure, etc., at a regular time interval. As a consequence, the applicative software is mainly composed

of hard deadline constrained tasks which execute repetitively with fixed periods. Additionally, a real-time system may have aperiodic tasks with random arrival times and no deadline. Thus, in the so-called HRT (Hard Real-Time) applications such as medical critical care, the RTOS has to guarantee that all hard deadlines are met and to provide minimal responsiveness to the aperiodic tasks, executing them as soon as possible [31]. In contrast, a SRT (Soft Real-Time) application, such as animal tracking, tolerates deadline missing from the operating system [32].

Today's classical real-time schedulers, including RM (Rate Monotonic) and EDF (Earliest Deadline First) [33], fail in energy-harvesting systems where the supply energy is intermittent. Energy and time should be treated as equally important resources. A number of problems such as intermittent computing, dynamic power management and energy aware real-time scheduling are not addressed in the design of a classical battery-powered sensor [34–38]. In particular, we have now to characterize the tasks by both processing time and energy consumption. Moreover, we have to achieve the online monitoring of available energy in the storage unit as well as the online prediction of environmental energy produced in the near future.

Consequently, the selection of an appropriate scheduling algorithm becomes necessary so as to guarantee energy neutrality of the RTEH sensors by considering both time and energy in the on-line decisions of the scheduler. The RTOS installed in any energy-neutral device has to treat energy as a central constraint, identically to timeliness. Energy may be more important than timeliness in some time intervals where not enough energy can be harvested so as to execute all the deadline constrained tasks in a timely manner. In order to deal with such a faulty situation, the scheduler should deliberately switch the processing unit to the sleep mode and postpone the execution of the currently active task, thus avoiding energy starvation for a future occurring task with high criticality. This is what is known as intermittent computing. A new energy aware real-time scheduling framework called ED-H, described in [39], combines the conventional dynamic priority-based algorithm EDF with intermittent computing facilities.

This paper introduces the scheduling issue in energy-neutral sensors and gives a short survey of the fundamental results about EDF-based scheduling for energy-neutral sensors.

The remainder of this paper is structured as follows: The subsequent section details relevant works performed on scheduling algorithms for real-time energy-neutral devices. Assumptions of the system model are presented in Section 3. In Section 4, we discuss the challenges for an optimal scheduling algorithm. This section also shows that the famous EDF scheduler has interesting properties that make it a good candidate for scheduling tasks when the ambient harvested energy is not predictable. A description of the optimal scheduler ED-H is given in Section 5. The schedulability analysis is reported in Section 6. Section 7 addresses the problem of aperiodic task servicing and describes an optimal slack stealing server for energy-neutral systems. Section 8 is about the considerations for implementing energy-neutral systems within the Operating System. Finally, Section 9 concludes the paper.

## 2. Related Works

Scheduling periodic tasks is a central issue in real-time embedded systems. Real-time scheduling typically focuses on models where the tasks have to be completed before a deadline and have processing requirements only. The work of Liu and Layland reported in [33] five decades ago deals with fixed-priority and dynamic-priority task scheduling. There, all the tasks execute cyclically and do not synchronize. A lot of works have considered this model, where there is no limitation on energy availability. Surveys can be found in [40,41]. Another important issue concerns real-time systems that consist of both aperiodic and periodic tasks. Aperiodic tasks have irregular arrival times and no deadline. The objective of the aperiodic task server is to minimize the response times for aperiodic tasks and guarantee hard deadlines for periodic tasks. Different approaches were proposed, including the Polling server, Deferrable server [42], Constant Bandwidth server (CBS),

Total Bandwidth server (TBS) [43], and Slack Stealing server, also known as EDL (Earliest Deadline as Late) [44,45].

The scheduling issue in energy-neutral devices has been explored from the beginning of the 2000s. The work of Allavena and Mossé, reported in [46], addresses frame-based tasks with voltage and frequency scaling capabilities. An optimal scheduler is presented, under the assumption that the energy storage unit replenishes with a constant power rate. Another work considers tasks with different software versions in [47]. An optimal EDF-based algorithm known as Lazy Scheduling Algorithm (LSA) is proposed by Moser et al. [48]. The harvested power is modeled as a time varying variable and all tasks consume energy with the same rate. LSA is extended by Liu et al. to apply to DVFS processors [49,50]. The so-called EA-DVFS and HA-DVFS algorithms take advantage of the slack time to slow down the processor and consequently to save energy. They speed up the task execution in case of the overflowing harvested energy. In [51], Abdeddaim et al. propose a fixed-priority scheduling algorithm, called $PFP_{ASAP}$. Periodic tasks consume energy linearly and the environmental source delivers constant power. A schedulability condition is given for $PFP_{ASAP}$.

Very recently, some works concentrate on semi-HRT applications that accept missed deadlines because of energy starvation. In [52], the incoming source energy is assumed to be predicted with accuracy. The objective of the best effort scheduler is to maximize the performance. The problem of the resilient scheduling against the energy-harvesting rate prediction error is studied in [53]. An energy-resilient scheduler is proposed for periodic tasks with multiple performance levels. In this work, the actual harvesting rate in each hyperperiod (equal to the least common multiple of the task periods) is considered as a constant value. The proposed scheduler allows one to react to unpredicted changes called surprises when the actual amount of harvested energy is different from the prediction. Simulations show that the scheduler outperforms other ones since it recovers from changes in a timely manner, adequately controls the performance degradation, and consequently makes the system survivable.

### 3. The System Model

#### 3.1. Assumptions about the Computing Load

We consider an energy autonomous system as depicted on Figure 2. Applications executed on wireless sensors have real-time constraints. Every application can be characterized by a generic set of periodic tasks, say $\tau$, which has $n$ deadline-constrained tasks. The tasks are independent, i.e., they do not synchronize. The tuple $(C_i, D_i, T_i, E_i)$ defines task $\tau_i$ with $C_i$, its worst-case computation requirement, $D_i$, its relative deadline, $T_i$, its repetition period, and $E_i$, its worst-case energy requirement. The values $C_i$ and $E_i$ can be derived from a static program analysis. The common assumption is that $0 < C_i \leq D_i \leq T_i$. $\tau_i$ generates an infinite number of instances called jobs. The ratio of the processing time spent in executing $\tau$ provides the so-called processor utilization factor, i.e., $U_{pp} = \sum_{\tau_i \epsilon \tau} \frac{C_i}{T_i}$. The average power consumed by $\tau$ is called the energy utilization factor and is given by $U_{ep} = \sum_{\tau_i \epsilon \tau} \frac{E_i}{T_i}$. The processing unit, generally a microcontroller, has one operating frequency. Its energy consumption comes from the dynamic switching activity of the circuits. The overhead due to switching the processor from one job to another is assumed to be negligible.

#### 3.2. Assumptions about Energy Production

This paper focuses on an energy-neutral system, which draws energy from environmental sources such as sunlight, vibrations, movement, etc. The system uses an energy harvester that generates electric power, which may be modeled with the function $P_p(t)$. The instantaneous charging rate $P_p(t)$ incorporates all losses due to friction, heat, air damping, etc. It follows that the amount of harvested energy during the time interval $[t_1, t_2)$ is given by the following formula: $E_p(t_1, t_2) = \int_{t_1}^{t_2} P_p(t)dt$.

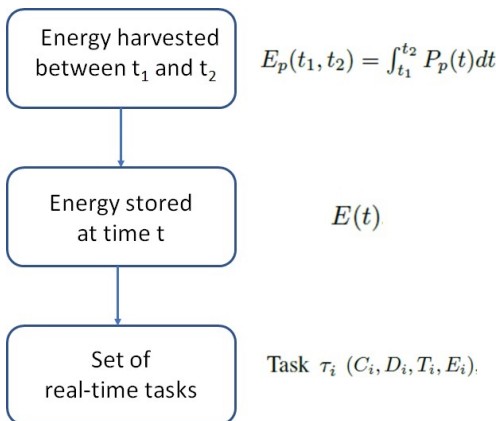

**Figure 2.** The energy model of the considered autonomous system.

*3.3. Assumptions about Energy Storage*

A battery and/or a capacitor are generally used to buffer the harvested energy. The storage unit then permits one to guarantee the operation of the wireless device for a certain period of time when the environment does not produce energy. The model assumes that energy consumption may overlap with energy production and power produced at any time instant never exceeds power consumed at the same time. Consequently, the residual capacity of the energy storage decreases every time some task executes on the computing unit.

The model considers an ideal energy storage with no leakage, which may be operating even in the absence of energy to harvest, provided the storage is not completely depleted. The capacity of the energy storage is denoted by $C$, which gives the highest amount of energy that can be stored at any instant. The quantity of energy stored at a given time $t$ is denoted by $E(t)$. The energy storage unit stops drawing power from the harvester when fully charged.

## 4. Challenges for Optimal Scheduling

*4.1. Necessary Terminology*

Below, we first provide some background on the terminology around energy neutral systems.

- *Optimality*: a scheduling algorithm is said to be optimal whenever each time a task set can be scheduled to meet its timing requirements on a given hardware platform; the same task set is feasibly scheduled by the optimal algorithm on the same hardware platform. Let us note that the energy harvester, the energy storage unit, and the computing unit characterize any platform.
- *Clairvoyance*: A clairvoyant scheduling algorithm has a precise knowledge of future arriving jobs and future energy produced by the source.
- *Lookahead-ld*: a scheduling algorithm is said to be a lookahead-ld if it needs to foresee on a time interval with a length equal to ld time units.
- *Idling*: an idling scheduling algorithm may keep the processor in the sleep mode even if there are one or more jobs ready for execution. In contrast, a *non-idling or work-conserving* algorithm such as the classical EDF and RM schedulers executes tasks as soon as possible, never inserting idle times in the schedule.

We now detail and explain important research results.

*4.2. No Optimality without Clairvoyance*

The following theorem says that any optimal energy aware scheduler requires clairvoyance on the future so as to anticipate energy starvation. Finding a valid schedule with no deadline missing whenever one exists cannot be possible by a totally on-line scheduling algorithm.

**Theorem 1.** *No optimal non-clairvoyant algorithm exists for scheduling tasks on a single processing self-powered sensor [54].*

Theorem 2 states that any faulty situation such as the energy depletion conducted for the missing deadline has to be anticipated early enough by the scheduling algorithm to make the right decision. More precisely, a lower bound on the clairvoyance interval is given by the longest relative deadline of the application. As a consequence, the longest deadline appears as a central parameter: the longer the relative task deadline is, the more long should be the time interval for predicting the harvested energy. If the value of the deadline is revealed to be much greater than the capability of the energy predictor, a sub-optimal schedule will be constructed. When designing a self-powered sensor, the central challenge consists in addressing the prediction issue of the future harvested energy. When there is no possibility to predict the incoming energy even in the short term, only a sub-optimal solution may be obtained using a simple non-clairvoyant online scheduler such as EDF.

**Theorem 2.** *Let D be the longest relative task deadline of the application. A lookahead-ld scheduling algorithm may be optimal only if $ld \geq D$ [54].*

*4.3. Scheduling with EDF for Unpredictable Systems*

The well known Earliest Deadline First (EDF) scheduler is optimal [33] whenever the energy of the tasks can be consumed greedily. The optimality signifies that EDF can successfully schedule any set of periodic tasks with deadlines if at least one valid schedule exists for the task set.

EDF makes processing decisions without any knowledge on, first, the arrival pattern of future tasks and second, the amount of harvested energy in the short-term future. From the previous theorems, EDF may only provide a sub-optimal solution to the scheduling issue under the energy-harvesting settings.

Nonetheless, in some applications, the energy source may be both uncontrollable and unstable. Recall that the optimal scheduler needs to adapt the activity of the processor in its dependence on energy availability, then define the so-called intermittent computing. Some tasks have to be postponed if the available energy in the storage unit is not sufficient to execute them completely or if executing them immediately will provoke energy starvation in the future. The optimal scheduler does not execute the tasks in ASAP (As Soon As Possible) mode. Intermittent computing is achieved through a smart dynamic power management strategy that decides when and for how long the processor should stay idle. This assumes that the system has all the necessary hardware and software for precisely determining how much energy is left in the energy reservoir (e.g., battery or supercapacitors) and for predicting the future incoming energy. Consequently, there is no chance to build an optimal schedule when the incoming energy is stochastic and uncontrollable. Furthermore, it appears of primary importance to assess the performance of the basic EDF strategy when no prediction mechanism is made possible. Theorem 3 then states that the classical EDF strategy remains the best scheduler.

**Theorem 3.** *EDF is the best non-idling strategy for scheduling tasks on a single processing self-powered sensor [55].*

The analysis of EDF relating to its competitive ratio (definition of competitive ratio is given in [56]) establishes a performance bound of EDF for the worst-case scenario in comparison to any optimal clairvoyant algorithm. Then, Theorem 4 enables us to state that unfortunately, EDF is non competitive. In other terms, it has a competitive factor equal to 0.

**Theorem 4.** *EDF is non competitive for scheduling tasks on a single processing self-powered sensor [55].*

As a consequence of Theorem 4, EDF may behave very poorly even if proved to be the best non-idling scheduler (see Figure 3). Nonetheless, let us note that a non-idling scheduling presents the following important features: easy implementation and low run-time overhead of the scheduler.

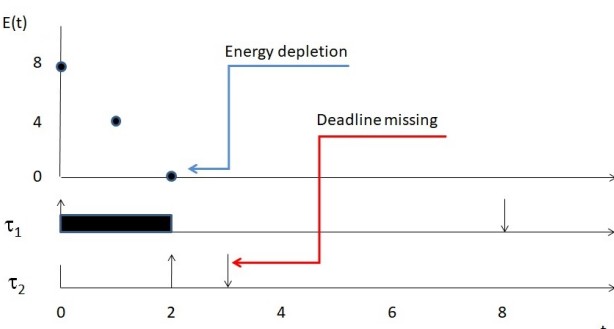

**Figure 3.** The EDF scheduler under energy-harvesting constraints; energy depletion at $t = 2$ leading to deadline violation at $t = 3$.

## 5. Optimal Scheduling under Energy-Harvesting Settings

In this section, we assume that the ambient energy may be accurately predicted in the near future, generating a negligible loss time and energy cost. We next demonstrate that taking advantage of lookahead permits to make EDF an energy-harvesting-aware scheduler of the system has intermittent computing facilities.

### 5.1. Principles of the ED-H Scheduler

The ED-H scheduling algorithm must choose between stopping the processor or selecting the task with the highest priority to execute. Starting the execution of a task as soon as possible will maximize the chance of a timely execution at or before the deadline [44]. However, the execution of a task should not compromise the timely execution of any other task that will arrive in the future. As a consequence, any decision should be based on the precise knowledge of both the profile of the future incoming energy, the current amount of energy available in the storage unit, and the amount of energy required by very urgent tasks that will occur in the future.

ED-H bases its scheduling decisions on the relative urgency of the ready tasks at any current time. The next task to be run has the closest deadline. In contrast, ED-H [39] has a work-conservative behavior, i.e., it may decide to put the processor in sleep mode deliberately even if some tasks are waiting for execution. ED-H authorizes immediate execution of a task only if this decision guarantees that no energy starvation will occur in the furure. Let us define the *preemption slack energy* at current time $t_c$ as the largest amount of energy that could be consumed continuously from $t_c$ by the currently active task, preventing energy starvation. In other terms, ED-H imposes processor idleness if the preemption slack energy is zero. In summary, ED-H forces the processor to be busy in one of the two situations: either the level in the energy storage unit has fallen below a threshold or the preemption slack energy has reached zero. In contrast, ED-H forces the processor to be idle in the two situations: either the energy storage unit is fully charged and no energy could be wasted, or executing no task would lead at least one task to miss its deadline, i.e., the processor has no slack time. In the situation where the storage unit is neither fully charged nor completely discharged and the system has both slack time and slack energy, the user can decide to make the processor either active or passive.

The rule that specifies when to start and when to stop recharging the energy storage unit will determine a particular version of the ED-H scheduler. One of these versions, called ASAP-ALAP, consists of finishing the recharging phase only when the storage unit is fully charged and finishing the discharging phase when the energy storage unit is completely exhausted. Under such a version of ED-H, the electronic circuits spend a maximum continuous time in a low-power mode. Energy overheads as well as time

overheads issued from switching between the different power modes are thus minimized. Figure 4 depicts the functioning of the ED-H scheduler.

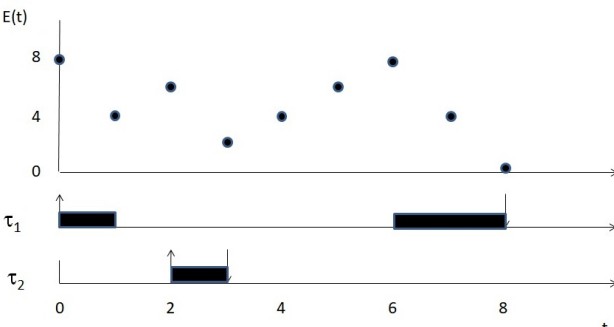

**Figure 4.** Illustration of the ED-H scheduler.

### 5.2. Implementation Considerations

Let us denote by $L_r(t_c)$ the ordered set of tasks which are ready at the current time $t_c$. The ED-H algorithm uses the following rules:

- **Rule 1:** The EDF-based priority assignment rule selects the future active job from $L_r(t_c)$.
- **Rule 2:** If $L_r(t_c) = \varnothing$, the processor is put in the idle mode on $[t_c, t_c + 1)$.
- **Rule 3:** If $L_r(t_c) \neq \varnothing$ and either $E(t_c) \approx 0$ or $PSE_{\mathcal{T}}(t_c) \approx 0$, the processor is put in the idle mode on $[t_c, t_c + 1)$.
- **Rule 4:** If $L_r(t_c) \neq \varnothing$ and either $E(t_c) \approx C$ or $ST_{\mathcal{T}}(t_c) = 0$, the processor is put in the busy mode on $[t_c, t_c + 1)$.
- **Rule 5:** If $L_r(t_c) \neq \varnothing$, $0 < E(t_c) < C$, $ST_{\mathcal{T}}(t_c) > 0$ and $PSE_{\mathcal{T}}(t_c) > 0$, the processor can be put either in the idle mode or in the busy mode.

### 5.3. Performance Analysis

Theorem 5 says that if a set of deadline constrained tasks is schedulable by any algorithm on a platform composed of a given processor, energy harvester, and energy reservoir, then it will be schedulable on the same platform using the ED-H algorithm.

**Theorem 5.** *The ED-H algorithm is optimal for scheduling tasks on a single processing self-powered sensor [39].*

Theorem 6 says that any set of the deadline-constrained tasks, which are schedulable by ED-H using a given energy harvester, will still be schedulable using a more powerful harvester. Consequently, we may perform the feasibility analysis under a worst-case scenario in which the energy harvester delivers a constant power even if it is lower than the actual one. The challenge lies in the determination of an accurate and constant lower bound on the environmental power. The finer the approximation of the source power, the more exact is the schedulability test.

**Theorem 6.** *The ED-H algorithm is robust with respect to source power [57].*

Power harvested from the environmental source may exhibit stochastic fluctuations at runtime. Some energy sources make it impossible to compute an acceptable lower bound, off-line. In that situation, only an online admission test will be achieved so as to repetitively test the schedulability based on the approximate and constant harvested power computed over time by the energy predictor, thus reducing both the run-time overhead and memory utilization.

## 6. Schedulability Testing

The schedulability analysis is the central part of the real-time scheduling issue. In systems with no energy limitation, the processor demand approach permits one to detect the

processing overload. More precisely, the technique consists of verifying that the duration of each time interval always exceeds the amount of processing time required in that interval. The idea for analyzing the schedulability with ED-H is to follow a similar approach in the energy domain. In other terms, we have to verify that in any time interval, the available energy exceeds the amount of energy required by the tasks for their timely execution.

Consider the feasibility problem for a periodic task set that have to execute in a timely manner on a self-powered sensor. Since ED-H is optimal, this scheduling algorithm can be applied to provide a necessary and sufficient feasibility test. As any set of periodic tasks produces exactly one collection of jobs, the feasibility problem concerns the verification that this collection of jobs is scheduled in a feasible way to meet all the deadlines under ED-H.

### 6.1. Schedulability Testing of a Generic Job Set

Here, the feasibility decision problem is addressed for a generic set of jobs $J = \{J_1, J_2, \ldots, J_n\}$, issued or not from periodic tasks [39]. The job $J_i$ is completely specified by the four-tuple $(r_i, C_i, E_i, d_i)$. It respectively provides the date of arrival, called the release time, worst case execution time (expressed in time units and normalized to the processing capacity), worst case energy consumption (expressed in energy units), and deadline of $J_i$.

The factors which limit schedulability may be the processing time or/and energy. As a consequence, and additionally to analyze the processor demand as in a real-time scheduling theory under no energy limitations, the energy demand analysis has to be considered. This leads us to define two situations, which are called time starvation and energy starvation:

- *Time starvation*: situation when the amount of processing time required by a job until the deadline is not sufficient, while enough energy is available when the deadline missing occurs.
- *Energy starvation*: situation when the amount of processing time required by a job until the deadline is sufficient but the energy is exhausted when the deadline missing occurs.

Time schedulability testing aims to check the absence of the time starvation, while the energy schedulability testing checks the absence of energy starvation in any time interval.

Let us introduce the static slack time of the job set $J$. The processor demand of the job set $J$ on the time interval $[t_1, t_2)$ is given by $h(t_1, t_2) = \sum_{t_1 \leq r_k, d_k \leq t_2} C_k$. We define the static slack time of $J$ on $[t_1, t_2)$ as $SST_\tau(t_1, t_2) = t_2 - t_1 - h(t_1, t_2)$. $SST_J(t_1, t_2)$ represents the longest time that could be made available within $[t_1, t_2)$ after having executed all the jobs of $J$ with a release time at or after $t_1$ and a deadline at or before $t_2$. Finally, the static slack time of $J$ is given by

$$SST_J = \min_{0 \leq t_1 < t_2 \leq d_{Max}} SST_J(t_1, t_2). \tag{1}$$

In the same manner, we may introduce the static slack energy of the job set $J$. The energy demand of $J$ on the time interval $[t_1, t_2)$ is obtained by $g(t_1, t_2) = \sum_{t_1 \leq r_k, d_k \leq t_2} E_k$. Let $E_p(t_1, t_2)$ be the amount of energy which is produced by the source between $t_1$ and $t_2$. The static slack energy of $J$ on the time interval $[t_1, t_2)$ is given by $SSE_J(t_1, t_2) = C + E_p(t_1, t_2) - g(t_1, t_2)$. $SSE_J(t_1, t_2)$ represents the largest quantity of energy that could be made available within $[t_1, t_2)$ after having executed all the jobs of $J$ with a release time at or after $t_1$ and a deadline at or before $t_2$. Finally, the static slack energy of $J$ is given by

$$SSE_J = \min_{0 \leq t_1 < t_2 \leq d_{Max}} SSE_J(t_1, t_2) \tag{2}$$

Intuitively, the static slack time is a lower bound on the acceptable processing surplus at any instant. The static slack energy of $J$ gives a lower bound on additional energy, which could be consumed at any instant. The following theorem stated in [39] can be derived:

**Theorem 7.** *$J$ is schedulable by ED-H if and only if*

$$SST_J \geq 0 \; and \; SSE_J \geq 0 \tag{3}$$

Theorem 7 says that Equation (3) guarantees that both energy and real-time requirements can be satisfied for the job set *J*. If at least one energy- and time-feasible schedule exists for *J*, the optimal ED-H scheduler will build it. Note that the analysis presented in Theorem 7 adapts to any time varying operational condition.

*6.2. Schedulability Testing of a Periodic Task Set*

Many embedded system applications including self-powered sensor nodes in the IoT have different types of real-time constraints. Nonetheless, most of them implement hard periodic tasks dedicated to the controlling functions which execute cyclically and have critical deadlines. If one deadline is missed, this may have serious repercussions, notably in healthcare applications. On the other hand, soft real time tasks do not have critical deadlines and the main focus is to minimize their response time, i.e., the time between the task arrival and task completion (see Section 7). We are consequently interested in the schedulability check of a set of periodic tasks.

**Theorem 8.** *If a periodic task set $\tau$ is schedulable by the ED-H scheduler in the synchronous scenario, $\tau$ is also schedulable in any asynchronous scenario [57].*

The initial release times of the periodic tasks cannot necessarily be known in advance. Theorem 8 says that the worst-case is attained in the synchronous arrival sequence, i.e., when all the tasks release jobs at time 0 as in the classical hypothesis with no energy limitation. As a consequence, a sufficient feasibility condition for any periodic task set, synchronous or not, is equivalent to consider the worst-case synchronous scenario.

The question which results from Theorem 8 is *how to verify whether a synchronous periodic task set can be feasibly scheduled under energy harvesting settings.* The hyperperiod $H_\tau = LCM\{T_1, T_2, \ldots, T_n\}$ is an obvious time-bound for the testing schedulability with EDF when the energy is not limited. As proved in the following theorem, this time-bound still exists with ED-H under energy-harvesting considerations when the source power is constant over time.

**Theorem 9.** *Assume a constant source power. Any synchronous periodic task set $\tau$ is schedulable by ED-H if and only if ED-H produces a valid schedule in the time interval $[O, H_\tau)$ [57].*

Thanks to Theorem 9 and the robustness properties of the ED-H scheduler, the testing feasibility of a periodic task set can now be reduced to check that there is no energy shortage and no deadline missing in only one hyperperiod.

In many applications, the harvested power can be assumed as a constant on long time intervals, regarding periods of the tasks. For example, solar-powered sensors experience important changes over time in the power harvested because of the diurnal cycle in the sunlight, varying conditions of the weather, and seasonal patterns. However, these changes will not occur every second, whereas the period of a controlling task is in the order of a fraction of a second. Energy prediction models have been developed in order to estimate the expected energy intake in the near future [58–60]. Thus, we may apply an energy prediction technique so as to determine the constant harvested power for the next time interval with a duration in the order of multiple task hyperperiods. Let us provide the example of energy harvesting from the human body, which generates a constant power by using thermoelectric generators. Such wearable devices allow one to exploit the temperature differences between the environment and surface of the human body.

One other schedulability approach is based on the robustness properties of the system to implement off-line testing. The robustness refers to the capability of the system to guarantee a stable behavior when positive changes occur on the parameters (e.g., higher harvested power than expected, lower energy consumption than expected, etc.). Feasibility checking can be achieved off-line when the energy profile is precisely characterized for the application lifespan. If we consider the particular case of a constant power source, the schedulability problem is reduced to testing a simple sufficient condition. Even if

it is based on pessimistic assumptions, especially regarding the energy production, the feasibility is guaranteed for the entire lifetime of the application, which is suitable for hard real-time systems. Such an approach also has a practical relevance, mainly coming from a low computational complexity and easy integration in the design process, since it passes over the variability of the environmental source power and does not need any additional software in charge of the energy prediction.

## 7. SSP: Optimal Scheduling for Aperiodic Tasks

### 7.1. The Scheduling Issue

Periodic and soft aperiodic jobs are scheduled together in most hard real-time task systems. The goal of task scheduling is to minimize the aperiodic responsiveness while still fulfilling the periodic task deadlines. A soft aperiodic task never causes serious damage even if it has a very long response time. Slack stealing is a method that overcomes the drawbacks of the background servicing. It provides minimal responsiveness for aperiodic tasks [44,45] by executing the periodic tasks in an ALAP mode, when at least one aperiodic task is waiting to be processed. In other words, any available processing time is made disposable for the aperiodic ones as quickly as possible. When no aperiodic task requires execution, the periodic tasks execute in the ASAP mode according to the EDF scheduling algorithm. We show hereafter how the slack stealing approach may be extended to RTEH systems with additional energy constraints.

### 7.2. Optimal Responsiveness with SSP

We submit here a slack stealing server that expands the original one to EH settings because it is tailored to the needs of an energy-neutral system. The so-called SSP (Slack Stealing with Energy Preserving) server authorizes aperiodic task executions as long as they do not violate any deadlines. Let us recall that a deadline breach is caused by either a shortage of the processing time or a deficiency of energy.

The slack of the periodic task set $\tau$ at the current time $t_c$ is evaluated as a pair of values by the SSP server. The first one estimates the slack time of $\tau$, which is defined as the excess of the maximum continuous processing time available for completing new jobs from $t_c$. The second one is the slack energy of $\tau$, which is primarily determined as the amount of energy surplus that the system may consume continuously from $t_c$. Periodic tasks are performed using ED-H when no aperiodic tasks have appeared. The server exploits the gathered slack time and slack energy to handle aperiodic tasks as soon as at least one aperiodic task arrives. When the aperiodic queue is not empty, the slack stealer is ready to execute. When there is slack, such as slack time or slack energy, the slack stealer is afforded the first priority. When there is no slack time or energy, it is given the lowest priority. The aperiodic task is determined by the slack stealer in the FCFS order.

The pseudo-code Algorithm 1 describes the framework of the SSP server and Figure 5 illustrates an example of its application.

The SSP task server is proved optimal in terms of aperiodic responsiveness:

**Theorem 10.** *Assume a set of periodic tasks feasibly scheduled by ED-H and a stream of occurring aperiodic tasks served in the FCFS order. The Slack Stealing server minimizes the response time of every aperiodic task [61].*

### 7.3. Simulation Results

The following simulation study is conducted to verify the theoretical performance of the SSP aperiodic task server in comparison to classical techniques. The central performance metric is the average response time of the aperiodic tasks since we want to optimize the responsiveness of the aperiodic tasks. The goals in this simulation experiment are first to measure the performance of SSP over a large set of tasks, and second to compare the performance against known algorithms.

---

**Algorithm 1** SSP: Slack Stealing server for ED-H

---

  **while** True **do**
    **if** $A_r(t)$ is not empty **then**
      calculate $ST(t)$ and $SE(t)$
      **if** $ST(t) > 0$ and $SE(t) > 0$ **then**
        execute Slack Stealer
      **else**
        execute ED-H
      **end if**
    **else**
      execute ED-H
    **end if**
  **end while**

---

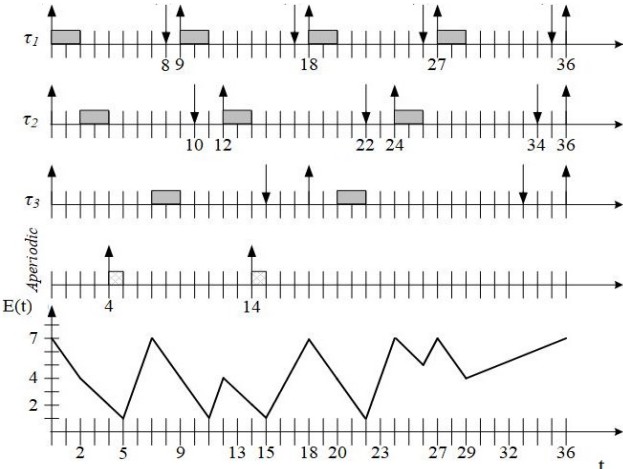

**Figure 5.** Illustration of the Slack Stealing server.

The algorithms used for comparison are the Background with Energy Surplus (BES) and Background with Energy Preserving (BEP). When there are no periodic activities to execute and the energy storage is depleted, BES services the aperiodic jobs. Any aperiodic task may be executed under BEP only if it does not cause an energy paucity for a periodic task released in the future. A Matlab simulation study was carried out. Each plot in the graphs represents 100 simulation runs, which are made up of one task set with 20 periodic tasks scheduled on 10 hyperperiods. The overall processing load $U_p$ includes 50% of the periodic processor utilization $U_{pp}$ and 50% of the aperiodic processor utilization $U_{ps}$. The overall energy consumption $U_e$ is made up of 50% of the periodic energy utilization $U_{ep}$ and 50% of the aperiodic energy utilization $U_{es}$. We postulate that the recharging power $P_p$ does not change over time in our study. A Poisson arrival pattern is used to generate a stream of aperiodic tasks with a uniform distribution.

Figure 6 reports the average response time of the aperiodic tasks for SSP, BES and BEP algorithms, respectively, with the processor utilization ratio sweeping from 0.1 to 0.9, while the average power consumed by the tasks is set to 80% of the power produced by the source. From $U_p = 0.5$ until $U_p = 1$, the Slack Stealing server outperforms the Background algorithms. In comparison to the Background curves, it has a normalized response time that is decreased by more than 15%. BEP is 1.5 times better than BES when executing aperiodic tasks. When $U_p$ raises, it smoothly concurs with BES. Aperiodic tasks are completed as long as there is always enough energy for upcoming periodic activities. Despite the increased energy constraint, the SSP can still cause a substantial reduction in aperiodic responsiveness compared to the Background services. SSP has a response time that is at least 30 percent faster than BES and BEP when $U_p$ varies.

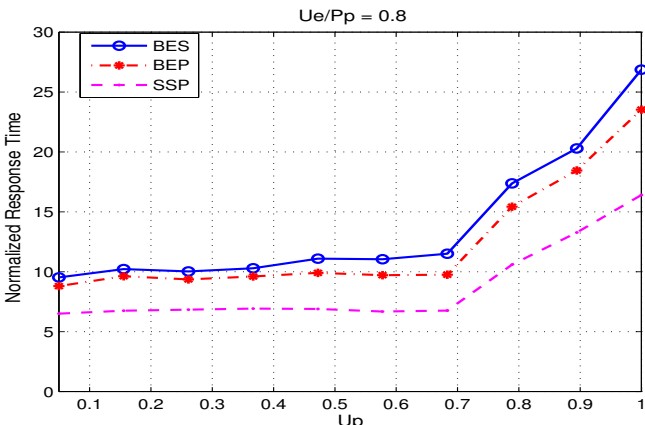

**Figure 6.** Response time by varying the processor utilization.

Figure 7 reports the average response time for SSP, BES and BEP algorithms, respectively, with the average consumption power sweeping from 0 to 1, while the processor utilization is 0.6. For all configuration parameters, the SSP server beats the two background strategies BES and BEP, as expected. It is worth noting that the higher the energy constraint, the better SSP performs compared to the background approaches. Because aperiodic tasks may only execute when the reservoir is completely refilled, BES performs worse for large energy requirements. When renewable incoming energy is abundant relative to the energy requirements, the BES and BEP produce similar results.

The SSP server attempts to harness time slack stealing to improve the CPU utilization and outperforms background servers significantly. Even when there is no energy restriction, they both act badly. The performance of the slack stealing-based server resembles that of the background servers when the system is severely time and energy confined.

Figure 8 depicts the overhead, which is the processing time the kernel spends conducting a service on behalf of an assigned task such as computing dynamic variables each time the scheduler is summoned. The on-line computing/updating slack time and slack energy are two factors of overhead in the SSP server. The frequency of the required computations will increase with the strength of time and energy limitations. It should be noted that the slack time and slack energy are computed whenever a new aperiodic task appears. In addition, in the absence of the aperiodic tasks, the slack time is computed promptly before subrogating in the sleep mode to recharge the energy storage unit.

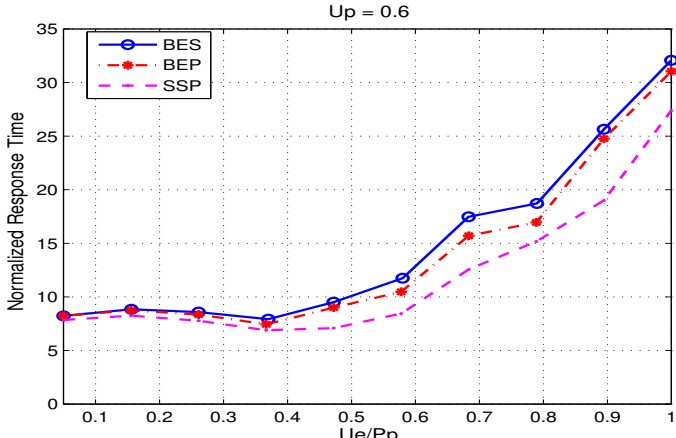

**Figure 7.** Response time by varying the average consumption power.

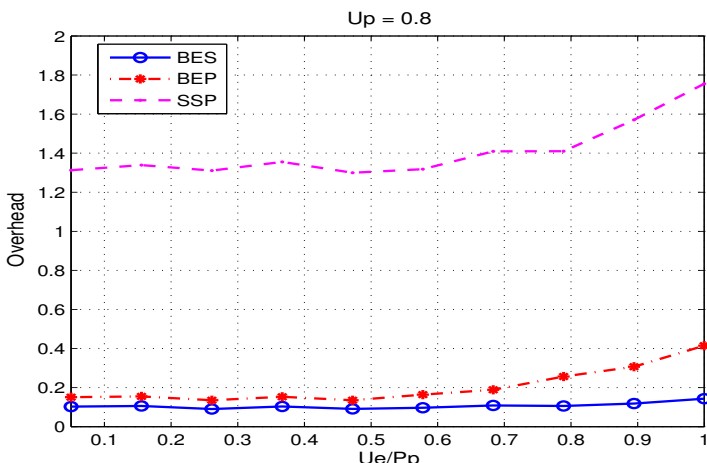

**Figure 8.** Overhead with Up = 0.8.

## 8. Implementation Considerations with Operating System

The most significant distinction between energy-neutral and traditional battery-operated devices is how they behave when their energy storage is depleted. An energy-neutral system goes into sleep mode at this moment and wakes up once enough energy has been harvested to continue the task execution. In contrast, a classical battery-operated system reaches the end of its lifetime. An energy-neutral system must be able to reliably store its state in permanent memory after detecting that there is no more energy in the storage. Additionally, the system must have detailed knowledge about task energy requirements, here called WCEC (Worst Case Energy Consumption) [62]. The system must have specified the minimum energy that should be available in the storage to authorize the wake up and guarantee the system to execute tasks for at least a given amount of time.

The Operating System has to obtain accurate and timely information so as to trigger energy-related events. The online monitoring of the state of the energy storage serves to notify the OS when a specified threshold is reached. Such functionality is essential, as it enables one to adjust the dynamic power management according to the amount of energy actually available. In addition, the system should be equipped with an energy prediction mechanism to avoid any future energy starvations. The slack energy of the system represents the highest amount of energy that could be continuously consumed by any task in execution. As a consequence, the computation of the slack energy is required at least before starting the execution of every task and possibly at regular time instants during the execution. This implies calling for the energy predictor. If the slack energy falls below a certain threshold, the OS is notified and the system switches on the sleep mode so as to recharge the energy storage. At this instant, just before the system enters the sleep mode, sufficient energy should be left to save the system state in a non-volatile memory. Identically, the recharging phase terminates as soon as there is no more slack time or enough energy becomes available in the storage unit. Figure 9 shows RTOS components with energy-harvesting considerations.

TinyOS [63,64] implemented with the programming language nesC, Contiki [65] and FreeRTOS [66], are very famous RTOSes. They all offer preemptive priority-driven task schedulers, so as to develop the IoT applications [67]. However, such RTOSes, even if suitable for battery-operated sensors, do not yet integrate clairvoyant and idling schedulers such as ED-H to accommodate the energy neutrality.

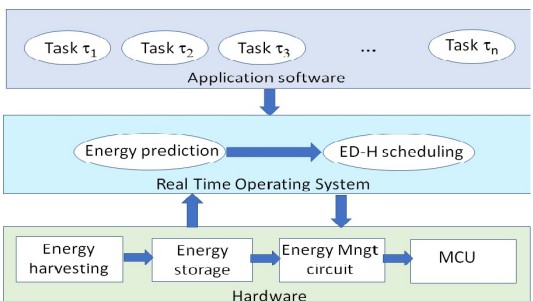

**Figure 9.** RTOS components with energy-harvesting considerations.

## 9. Conclusions

In a variety of application fields, self-powered wireless monitoring and control systems will be widely deployed. Making these systems as stable as possible, despite the intermittent generation of the environmental energy required to power them, is a significant problem. Timeliness and energy neutrality are the two challenging properties of reliable and sustainable real-time systems that distinguish them from battery-operated computing systems.

The paper has reviewed two key schedulers for energy-neutral devices, namely ED-H and SSP. The ED-H scheduler provides an intermittent computing framework for energy neutrality with no wasted energy, no energy starvation and no deadline missing whenever possible. The algorithm SSP combined with ED-H provides an optimal responsiveness to the aperiodic tasks, while guaranteeing no deadline missing for periodic tasks. The server SSP overcomes the drawbacks issued from Background approaches by taking advantage of the slacks both in the time domain and the energy domain.

**Author Contributions:** Conceptualization, M.C. and R.E.O.; methodology, M.C. and R.E.O.; software, R.E.O.; formal analysis, M.C.; investigation, M.C. and R.E.O.; data curation, R.E.O.; writing—original draft preparation, M.C. and R.E.O.; writing—review and editing, M.C. and R.E.O.; project administration, M.C.; funding acquisition, M.C. All authors have read and agreed to the published version of the manuscript.

**Funding:** This work is partially supported by the company e-Cobot with a grant under reference 2019-01050.

**Institutional Review Board Statement:** Not applicable.

**Informed Consent Statement:** Not applicable.

**Data Availability Statement:** Not applicable.

**Conflicts of Interest:** The authors declare no conflict of interest.

## Abbreviations

The following abbreviations are used in this manuscript:

| | |
|---|---|
| RTEH | Real-Time Energy Harvesting |
| CPS | Cyber–Physical System |
| EDF | Earliest Deadline First |
| ED-H | Earliest Deadline First for energy Harvesting aware |
| RM | Rate Monotonic |
| ASAP | As Soon As Possible |
| OS | Operating System |
| RTOS | Real Time Operating System |
| EDL | Earliest Deadline as Late as possible |
| CBS | Constant Bandwidth Server |
| TBS | Total Bandwidth Server |
| SSP | Slack Stealing with energy Preserving |

| FCFS | First Come First Serve |
|------|------------------------|
| WCET | Worst Case Execution Time |
| WCEC | Worst Case Energy Consumption |
| ST | Slack Time |
| SE | Slack Energy |

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
