# Peer review of "Earliest Deadline First Scheduling for Real-Time Computing in Sustainable Sensors"

_sustainability, doi:10.3390/su15053972_

Round 1

Reviewer 1 Report

The paper follows an advanced approach to IoT, namely Quality-of-service (QoS) issues.   Nevertheless, there are a few open issues that are to be addressed:

1) Assumptions: The article is based on clairvoyant job scheduling.  In practice, the scheduler does not know when the job will arrive.  How is that hanbdled in practice?

2) Is the hard-real time assumption adequate? under which constraints?

3) Simulation settings: what kind of real-time operating system has been considered for the simulations?  with pre-emptive kernel or not?

4) The introduction makes a good job.  However, the following strongly related references should be included from the reviewer's point of view.

Tia-96 DOI: 10.1007/BF00357882, talking about slack stealing scheduling for hard real-time OS.

Kocian-22 DOI: 10.1109/JIOT.2022.3142324 following a graphical approach to finding QoS in IoT such as scheduling + response time prediction in (soft and hard) real-time IoT systems.

Author Response

Dear Reviewer,

Reviewer 2 Report

This manuscript is basically well-written and the findings can contribute to helping in energy problems and carbon footprint minimization.

The authors should make sure if after the sum signs tiet is a product or subscript in the expression of Upp and Uep in page 2. 

In Page 5 in "lookahead-ld" what does "ld" mean?

In Figure 3 and 4 the variables are in line with the scale so put the variables in parallel with the scale. Mention in the figure legends what the arrows indicate.

In page 9 what do rk, dk and dmax mean?

In Figure 5 what variable is on the x-axis?

What does response time mean exactly in varying the processor utilization and the average consumption power? 

Is it possible to integrate different energy harvesting modes (for example light and thermal energy) by minimizing the need for place to improve energy supply?

Author Response

Dear Reviewer,

Reviewer 3 Report

This paper mainly summarizes the development of energy harvesting optimal scheduling technology. The principle of energy harvesting technology, the challenges faced by optimization scheduling technology, ED-H algorithm explanation, the scheduling optimization of SSP algorithm for aperiodic tasks, and the application of the algorithm in the operating system are described successively.

The article is fluent in language, but there are the following problems. It is recommended to revise it substantially:

1. The format of the first paragraph of the article needs attention in line 222, 338, 353, 441, 466, 498, 548, etc.

2. The title of the article cannot reflect that this article is a comprehensive paper.

3. The article is mainly about principles. Although the article describes the development and optimization process of energy acquisition and optimization scheduling, which is developed from the improved EDH algorithm based on EDF to the ED-H-SSP algorithm. ED-H-SSP can well solve the problem of aperiodic task deadline and energy depletion. However, if this research method is regarded as a main line, the article lacks a horizontal comparison with other branch research methods, and the content of comparison of advantages and disadvantages between this method and other branch methods should be added.

4. This paper proves the advantages of the SSP algorithm based on EDH in energy acquisition and scheduling through simple experiments, but the method is too simple and has no support of relevant papers.

5. As a summary paper, there are too few references.

6. Most of the references are studies conducted five years ago, and few refer to the latest research in the past five years.

Author Response

Dear Reviewer,

Reviewer 4 Report

The paper considers the new types of sensor-nodes that are able to harvest energy by green and long lasting sources such as solar. The research point that the authors described appropriately is the unstable and sometimes unpredictable manner of providing energy for the processing unit in such sensor-nodes. Therefore, they argued that new scheduling algorithms for the processing unit are required to ensure (as much as possible) that all the task can be processed before meeting their deadlines. To this end, the paper is well-presented. However, I have two suggestions, first, Figure 1 can be improved, and second, the related works section can be expanded. 

Author Response

Dear Reviewer,

Round 2

Reviewer 1 Report

The authors considered the reviewer's suggestions.

Author Response

Dear Reviewer,

Reviewer 3 Report

The author made some amendments to the comments and answered the reviewers' questions. However, the following problems remain unsolved:

Point 1: The format of the first paragraph in lines 222, 338, 353, 441, 466, 498 and 548 of the original manuscript has not been modified.

Point 2: The number of references for papers is still too small, and the proportion of papers in recent five years is relatively low. It is suggested that the author should try to enrich the references of the paper, so that this review can make a complete summary of the research in this field in recent years.

Author Response

Dear Reviewer,

Round 3

Reviewer 3 Report

The author has revised the paper according to the requirements, which is very good! Congratulations on being accepted.